# Transcriptomal Insights of Heart Failure from Normality to Recovery

**DOI:** 10.3390/biom12050731

**Published:** 2022-05-23

**Authors:** Mohammed Quttainah, Vineesh Vimala Raveendran, Soad Saleh, Ranjit Parhar, Mansour Aljoufan, Narain Moorjani, Zohair Y. Al-Halees, Maie AlShahid, Kate S. Collison, Stephen Westaby, Futwan Al-Mohanna

**Affiliations:** 1Department of Cell Biology, King Faisal Specialist Hospital & Research Centre, Riyadh 11211, Saudi Arabia; quttainah@kfshrc.edu.sa (M.Q.); vraveendran@kfshrc.edu.sa (V.V.R.); ssaleh@kfshrc.edu.sa (S.S.); dr.parhar@gmail.com (R.P.); kate@kfshrc.edu.sa (K.S.C.); 2Heart Centre, King Faisal Specialist Hospital & Research Centre, Riyadh 11211, Saudi Arabia; maljufan@kfshrc.edu.sa (M.A.); alhalees@kfshrc.edu.sa (Z.Y.A.-H.); maie@kfshrc.edu.sa (M.A.); 3Department of Cardiothoracic Surgery, Papworth Hospital, University of Cambridge, Cambridge CB23 3RE, UK; narain.moorjani@papworth.nhs.uk; 4Oxford Heart Centre, John Radcliffe Hospital, Oxford OX9 3DU, UK; swestaby48@gmail.com

**Keywords:** heart failure, RNA-Seq, hypertrophy, cardiac recovery, dilatation

## Abstract

Current management of heart failure (HF) is centred on modulating the progression of symptoms and severity of left ventricular dysfunction. However, specific understandings of genetic and molecular targets are needed for more precise treatments. To attain a clearer picture of this, we studied transcriptome changes in a chronic progressive HF model. Fifteen sheep (*Ovis aries*) underwent supracoronary aortic banding using an inflatable cuff. Controlled and progressive induction of pressure overload in the LV was monitored by echocardiography. Endomyocardial biopsies were collected throughout the development of LV failure (LVF) and during the stage of recovery. RNA-seq data were analysed using the PANTHER database, Metascape, and DisGeNET to annotate the gene expression for functional ontologies. Echocardiography revealed distinct clinical differences between the progressive stages of hypertrophy, dilatation, and failure. A unique set of transcript expressions in each stage was identified, despite an overlap of gene expression. The removal of pressure overload allowed the LV to recover functionally. Compared to the control stage, there were a total of 256 genes significantly changed in their expression in failure, 210 genes in hypertrophy, and 73 genes in dilatation. Gene expression in the recovery stage was comparable with the control stage with a well-noted improvement in LV function. RNA-seq revealed the expression of genes in each stage that are not reported in cardiovascular pathology. We identified genes that may be potentially involved in the aetiology of progressive stages of HF, and that may provide future targets for its management.

## 1. Introduction

Heart failure (HF) is a multidimensional clinical syndrome noted for a diminished ability of the heart to pump and/or fill with blood. Regardless of its different aetiologies, HF adds a great impact on economies and on social and clinical services. Despite some promising studies presenting a drop in the statistical trend of HF [1], others have shown an increase in incidence and prevalence because of the steadily growing elderly population. Moreover, epidemiological studies continue to verify the increased morbidity and mortality due to HF [2].

Mainstay conservative/pharmacological treatments may prove unsuccessful in some instances of HF. Consequently, implantation of ventricular assist device or heart transplantation may be the only option left in end-stage intervention despite their limitations and complications [3]. Calculable recovery of cardiac function has been demonstrated in our previous comparative model [4]. Geometrical improvement of the LV function associated with reverse remodelling and recovery had defined blood biomarkers documented in dilated cardiomyopathy (DCM) patients receiving pharmacological and/or device therapy [4,5,6,7]. Other studies combining magnetic resonance imaging (MRI) with electrocardiogram (ECG) in DCM showed specific associative ECG and MRI patterns [8]. Such markers are significantly associated but not directly correlated with structural LV changes in myocardial remodelling. Some have illustrated histopathological studies of myocardial fibrosis and hypertrophy could be studied as predictors of LV reverse remodelling (LVRR) [9,10,11]. Still, such orthodox methods cannot be taken as steadfast pointers of the progressive manifestation of LVRR. Several studies have taken the molecular high-throughput approach to investigate myocardial gene expression both spatially and temporally in search of more accurate predictors of LVRR. Studies using cross-platform analysis on HF patients have established a strong association between LVRR and expression of transcripts of coding or non-coding RNA species [12,13].

Next-generation sequencing studies [14,15,16] have demonstrated differential gene expression between different cardiomyopathies [10] and HF patients before and after implantation of an LV assist device (LVAD) [17]. Studies on long term mechanical unloading of the LV have shown that reducing LV wall stress resulted in geometrical changes such as a reduction in the thickness and the diameter of the LV. These changes demonstrated an increase in endothelial function and stem cell recruitment with consequentially improved calcium handling within the cardiomyocytes and increased mitochondrial respiratory function. Such improvements were accompanied by a reduction in inflammatory response and apoptosis and leading the extracellular matrix to modify its cytoskeletal proteins reservoir and modulate the use of total collagen to promote matrix remodelling [18]. Thus, LVAD support induces significant changes in myocardial gene expression, as pre- and post-LVAD hearts demonstrate significantly distinct genomic footprints. DNA microarray technology could distinguish, in a blind manner, patients with different HF aetiologies [19].

Clinical longitudinal studies on the progressive changes leading to HF and its recovery are quite scarce and insufficient, and so are data on transcriptome analysis associated with such changes in LVRR [20]. Here, we used a non-ischemic DCM (NIDCM) ovine model to investigate the transcriptome of HF and recovery by prospective endomyocardial biopsies during the development of various stages of the disease. Progressive pressure overload transitioned the LV into the defined stages, namely, hypertrophy, dilatation, and failure with changes in LV ejection fraction (LVEF). Each progressive stage leading to HF has a unique set of genes that are differentially expressed. Enrichment analysis by the PANTHER and Metascape and disease gene association analysis using DisGeNET retrieved potential candidate genes, which may otherwise remain unnoticed in the pathogenesis of hypertrophy, dilatation, or HF. Our data indicate that certain genes involved in circadian rhythm are associated with HF. Thus RNA-seq transcriptome analysis provides a spectrum of data for many investigations that need to be further validated experimentally for their direct involvement in cardiac hypertrophy, dilation, and HF.

## 2. Materials and Methods

### 2.1. Clinical Methodology

Surgical procedures were in adherence to the Guide for the Care and Use of Laboratory Animals by the National Institute of Health and approved by Animal Care and Use Committee by King Faisal Specialist Hospital and Research Centre. The detailed procedure was described in the previous publication from our laboratory [4]. Briefly, 15 male *O. aries*, six months old and weighing 29–35 kg, were prepared and fasted with access to water. Animals were sedated and scrubbed following surgical aseptic technique. Following muscle relaxation, endotracheal intubation was secured under full anaesthesia and continuous ventilation. Local anaesthesia and analgesics were used to control intra- and postoperative pain [4].

#### 2.1.1. Aortic Banding Procedure

The aortic banding procedure is described in detail in our previous publication [4]. Briefly, after approaching the pericardium through a third left antero-lateral thoracotomy, an inflatable double-cuffed constriction band was placed around the ascending aorta above the level of the coronaries. The band encloses a catheter balloon with an inflatable port embedded subcutaneously. A gradual increase in aortic Doppler velocity from an average of 5.22 ± 1.07 mmHg in the first week post-banding to an average of 51.36 ± 6.95 mmHg at the failure stage (116.5 ± 2.7 weeks) was achieved by steady weekly inflation under echocardiographic monitoring. Upon reaching clinical and echocardiographic stages of failure, the aortic cuff was deflated with the continuation of echocardiographic monitoring during recovery. Thirteen out of the fifteen animals were followed throughout the study. Two animals developed an infection and were therefore excluded.

#### 2.1.2. Endomyocardial Biopsy

As discussed in our previously published work [4], 8 to 10 endo-myocardial biopsies were collected from each sheep before aortic banding and throughout the progression to heart failure and during the stage of recovery. We carried out several procedures during each new stage of the pathological progression of the left myocardium and its eventual regression. An average of 10 biopsy samples per procedure was collected. Each biopsy sample weighed as little as 0.7 mg and no more than 1.9 mg. Right carotid artery access is used to introduce disposable biopsy forceps by gently pushing through the aortic valve aperture. Once inside the left ventricle, the introduced forceps were positioned and anchored on the endomyocardial wall. With the aid of a C-arm X-ray guided fluoroscope, biopsy samples of the endomyocardial tissues were taken and snap-frozen in liquid nitrogen. Tissue biopsy samples were taken from all five stages and subsequently stored at −80 °C or placed in 3.7% buffered formaldehyde in PBS (*v*/*v*). At the end of the study, animals were euthanatized whilst under general anaesthesia.

#### 2.1.3. Echocardiography

The subjects were gently restrained manually and placed on the left lateral position on an operating table. Transthoracic two-dimensional (2D) imaging, including Doppler studies, was performed regularly throughout the length of the study. The LV dimensions and ejection fraction were measured by two-dimensional guided M-mode method. The peak pressure gradient (PG), aortic Doppler velocity (AoVel), and associated curves were obtained from the suprasternal position. Left ventricular internal diameter in systole and diastole (LVIDs and LVIDd) and wall dimensions, interventricular septum thickness (IVSd and IVSs), and posterior wall thickness (PWTd and PWTs) were routinely obtained throughout the experiment. This allowed the left ventricular mass index (LVMI) and fractional shortening (FS), as a measure of ventricular contractility, to be calculated as follows: LVM(g) = 1.04 [(LVID + PWT + IVS)3 − LVID3] − 13.6; BSA m 2 = 0.112 (weight in kg)2/3; LVMI g/m^2^ = LVM/BSA [4,20].

### 2.2. Total RNA Isolation

Total RNA samples were extracted from selected (3 control, 4 hypertrophy, 4 dilated, 3 failure, and 3 recovery) representative biopsies for each stage mentioned above using a Qiagen RNeasy Fibrous Tissue Mini Kit following the manufacturer’s instructions. Although we aimed for a longitudinal analysis of gene expression, the total RNA for each stage was not derived from the same sheep due to the sample size. However, the physiological and clinical data between the sheep were comparable at each stage to presume that the sampling was comparable. The concentration and overall quality of RNA samples were assessed by Nanodrop ND-100 spectrophotometer and Agilent 2100 Bioanalyzer. The quality and quantity of RNA samples used for RNA-seq analysis are provided as Appendix A.

### 2.3. RNA-seq Analysis

RNA-seq bioinformatics analysis was outsourced to BGI Genomics (Hong Kong) using the BGISEQ-500 platform, averagely generating about 31.22 million reads per sample. All the samples had greater than 99% clean reads after filtering out unknown and low-quality bases, as well as adapter-polluted reads. The clean reads were mapped to the reference genome of *Ovis aries* using HISAT(v2.2.1) [21]. The average mapping ratio with the reference genome was 95.23%, and the average mapping ratio with the genes was 92.78%. A total of 22,467 genes were detected. The uniformity of the mapping result for each sample suggests that the samples are comparable. For gene expression analysis, the clean reads were mapped to reference transcripts using Bowtie2 [22], and the gene expression levels for each sample were calculated with RNA-seq by the expectation–maximisation (RSEM) software package (v1.0.6) [23]. The Pearson correlation of gene expression between samples was calculated by the “cor” function in “R” package. Hierarchical clustering between all samples was analysed by hclust, and PCA analysis was carried out using princomp. The diagrams of cluster dendrogram and PCA were drawn in ggplot2. Validation of a few selected genes from RNA-seq data was performed by qRT-PCR. Briefly, the cDNAs were prepared using 0.5 µg total RNA (leftover after sending for RNA-seq) using a high-capacity cDNA synthesis kit from Applied Biosystems according to standard protocol. The gene-specific qRT-PCR was performed using SYBR green master mix in a 7500 Applied Biosystems real-time PCR machine. The fold changes were quantitated using the ΔΔ*CT* method. The sequence of primers used were as follows: *GAPDH*: Fwd- 5′-CAAAGTGGACATCGTTGCCA-3′; Rvs- 5′-TGGAAGATGGTGATGGCCTT-3′, *TPT1*: Fwd-5′-GGGGCTGCAGAACAAATCAA-3′; Rvs- 5′-CAGCCAGTTTTGACGACAGG-3′; *NR4A3*: Fwd- 5′-CATGAGTCGGGATAGCAGCAT-3′; Rvs- 5′-TTCCGCAGTAGGCTTTGAATG-3′; *TIMP1*: Fwd- 5′-TCACCACCTGCAGTTTTGTG-3′; Rvs- 5′-TCCTCACAACCAGCAGCATA-3′; *GADD45B*: Fwd 5′-AACTCTTGCTCCTGGAGACC-3′; Rvs- 5′-CAGATGCCATCACCGTTCAG-3′.

### 2.4. Differentially Expressed Gene (DEG) Calculation

From the BGI-derived data, we excluded the genes that were not expressed greater or equal to 1 fragment per kilo base of exon per million mapped fragments (FPKM) in all the biological replicates of each stage. For differentially expressed gene analysis, we kept all the genes in control that had FPKM values equal to or above 1, and a pairwise comparison was performed with genes in each experimental stage (hypertrophy through recovery) that is expressed above the FPKM value of 5. Differentially expressed genes (DEGs) were identified using the DEseq2 algorithm by comparing each group to control (hypertrophy vs. control; dilated vs. control; failure vs. control; and recovery vs. control). The DEseq2 method relies on the negative binomial distribution and was calculated as described [24].

DEGs in each stage were then annotated for pathways, biological processes, molecular functions, and protein class by the web-based PANTHER database (for parent GO term enrichment), and Metascape for detailed GO child term enrichment, pathways, and disease–gene association.

### 2.5. Functional Annotation Analysis by PANTHER Database and Metascape

The PANTHER (Protein ANalysis THrough Evolutionary Relationships) classifies the proteins and their genes on the basis of biological process, molecular function, PANTHER pathway, and protein class. We used version 15.0 of PANTHER for the classification of DEGs. The DEGs in each stage were then annotated for PANTHER pathways, biological processes, molecular functions, cellular component, and protein class by the web-based PANTHER database (for parent GO term enrichment).

Metascape is a web-based tool that simultaneously searches for functional annotation of genes in different databases: KEGG Pathway, GO Biological Processes, Reactome Gene Sets, Canonical Pathways, DisGeNET, and CORUM (http://metascape.org/gp/index.html, accessed for analysis on 16 August 2020). Metascape also processes the disease–gene association using the appropriate database. Since *Ovis aries* was not included as input and analysis species in the Metascape interphase, we used *H. sapience* as the reference gene database. All genes in the genome have were as the enrichment background. Terms with a *p*-value < 0.01, a minimum count of 3, and an enrichment factor >1.5 (the enrichment factor is the ratio between the observed counts and the counts expected by chance) were collected and grouped into clusters on the basis of their membership similarities. More specifically, *p*-values were calculated on the basis of the accumulative hypergeometric distribution, and q-values were calculated using the Benjamini–Hochberg procedure to account for multiple testing. Kappa scores were used as the similarity metric when performing hierarchical clustering on the enriched terms, and sub-trees with a similarity kappa score > 0.3 form a cluster.

### 2.6. Gene–Disease Association Analysis

Metascape has the inbuilt capability to employ the use of the DisGeNET platform that integrates an expert-curated database of human diseases and their related genes, and also includes data from animal disease models [25,26]. Terms with *p*-value < 0.01, a minimum count of 3, and enrichment factor >1.5 were collected, and on the basis of similarities, they were clustered. To find the genes in each disease stage that were not associated with cardiovascular disease, an independent search was conducted in DisGeNET, and the genes that were not associated with cardiovascular diseases were filtered.

### 2.7. Statistical Analysis

One-way analysis of variance (ANOVA) was used for echocardiographic analysis. Bonferroni-corrected or Duncan’s post hoc analysis of the ANOVA results was used for multiple comparisons. Graphs were prepared in GraphPad prism 6.0. Data are presented as means ± SEM. Statistical significance was considered when *p* < 0.05.

## 3. Results

### 3.1. Validation of the Model by Clinical Assessments

The clinical assessment demonstrated that aortic-banded sheep displayed the expected echocardiographic profiles associated with the changes in the LV physiology and dimensions throughout the disease process (Appendix A) [4].

### 3.2. Hierarchical Clustering and Principal Component Analysis

Employing single replicate poly-A RNA-sequencing on LV tissue samples, we studied the changes in the profile of gene expression in the myocardium under a controlled and consistently increasing pressure overload that takes a normal LV into the stages of hypertrophy, dilatation, and failure. Then, we studied whether the LV recovery following the removal of the constriction was accompanied by a normalisation of the gene expression pattern.

All the analysed samples were clustered hierarchically on the basis of the expression of all genes (the presence of transcripts), and the clustering reflects the relationship between the samples (Figure 1A). Samples H6, H7, and S1H (hypertrophy) are very similar in their gene expression pattern and thus they were clustered closely. While a dilated (D13) sample and a failure (F16) sample were very similar in their gene expression. One hypertrophy (H5) sample was closely related to the control (N4) sample in gene expression. Two dilated (D12 and D11) and two recovery (R8 and R10) samples also showed similarity in their transcript expression. Two controls (N1 and N3) were also similar to a recovery (R9) and a dilated sample (S9D). We used principal component analysis (PCA) to broadly understand how closely the gene expression levels between each sample were and visualised the clustering of samples for the most differentially expressed genes. Using the PC1 and PC2, some of the biological samples of different stages displayed some overlap in the differential gene expression pattern (Figure 1B).

### 3.3. Recovery Reverses the Gene Expression Profile Close to Control

The pairwise comparison of gene expression in each pathological stage to control is shown in Figure 2A–D. The number described in each MA plot is composed of mRNAs, lncRNAs, and miRNAs. Figure 2E shows the heat map of significant gene expression of each group in terms of FPKM. The highest number of differential gene expressions was observed in the failure stage. Predictably, some transcripts were significantly expressed only in a single stage, while others were significantly expressed in multiple stages during the development of the disease. In hypertrophy, a unique upregulation of 134 genes was observed, while 33 were downregulated. The dilated stage uniquely showed 19 genes upregulated, and 21 genes were downregulated. In the failure stage, 118 genes were uniquely upregulated, and 105 genes were downregulated. When the condition set that the FPKM of DEGs in the recovery stage was greater than 5, no genes were found to be significantly different in expression compared to the control. When we relaxed the FPKM cut-off of recovery transcripts to ≥ 1, we found that five genes were differentially regulated compared to the control. This shows that the gene expression was nearly normalised in the recovery stage compared to controls, illustrating the validity of this experimental model (Figure 2F).

Certain sets of DEGs were found significantly up-/downregulated in more than one stage. The hypertrophy and dilated stages had 23 commonly expressed genes. Conversely, in hypertrophy and failure stages, the number of common DEGs was 43. The number of significantly expressed DEGs common in dilated and failure was 17. Among these genes that are commonly expressed in any two different stages, seven were common to all three stages. Figure 2G–N depicts the log_2_fold change expression top 10 up-/downregulated unique DEGs and common DEGs in all stages. The whole list of significantly expressed DEGs in each stage is provided in Appendix A.

Gene expression data derived from RNA-seq were validated and confirmed by qRT-PCR amplification of TIMP1, TPT1, GADD45B, and NR4A3. Our selection of these genes for the validation was because they were highly expressed genes either unique in one stage or common to more than one stage. Due to the sample scarcity, we could only select four genes in addition to GAPDH for validation (Figure 3).

### 3.4. Functional Annotation and Enrichment of DEGs Using the PANTHER Database

The significantly expressed genes in each stage were subjected to functional classification and enrichment analysis using the PANTHER database for biological process (BP), molecular function (MF), cellular component (CC), PANTHER pathways, and protein class. Figure 4A–C depicts the enrichment of MF, BP, CC, and PANTHER protein class (PC) for hypertrophy, dilated, and failure stages. The recovery stage did not retrieve any significant functional annotation because there were not many genes differentially regulated between control and recovery. The cellular process was the most enriched GO term under the BP for all three stages. The most enriched MF was binding, followed by molecular function regulator in the hypertrophy, and catalytic activity in the dilated and failure stages. The enrichment of PANTHER protein class in hypertrophy revealed 25 genes that are of gene-specific transcriptional regulator class. In the dilated stage, seven genes enriched the cytoskeletal protein class. In the LV failure stage, the most enriched protein class was metabolite interconversion enzymes with 26 genes.

PANTHER pathway (Figure 4D) analysis showed a significant contrast between hypertrophy and failure compared to dilated stage. In hypertrophy, enrichment for PANTHER pathways retrieved nine pathways as significantly enriched. GnRH receptor signalling pathways enriched with eight genes provide the possibility of the activation of this pathway. It is already known that GnRH agonists used in androgen deprivation therapy in prostate cancer increase the risk of cardiovascular diseases [27]. Likewise, the gene-enriched GnRH pathway may be playing a role in the initiation and progression of hypertrophy of the heart. The other significantly enriched pathways in hypertrophy are PDGF signalling pathway (six genes), CCKR (cholecystokinin receptor) signalling pathway (six genes), angiogenesis (five genes), and inflammation by chemokine and cytokine signalling pathway (five genes). The most enriched pathway in the dilated stage was inflammation mediated by chemokine and cytokine signalling pathway (three genes), followed by integrin signalling pathway, GnRH receptor pathway, p53 pathway, and CCKR signalling (two genes in all). It may be possible that in the dilated stage, there may be a deprivation of protein synthesis, which can impair the function of LV. In the failure stage, the most enriched pathways are GnRH receptor pathway (seven genes) and CCKR signalling (seven genes), followed by angiogenesis (six genes) and the integrin signalling pathway (five genes).

### 3.5. Metascape Enrichment Analysis of DEGs Showed Unique Pathways and Biological Processes in Each Stage of Disease

#### 3.5.1. Hypertrophy

The 210 DEGs were annotated to GO terms in Metascape. The most significant BP found to be enriched in the hypertrophy stage was circadian rhythm. The second most significant enriched BP was blood vessel development. The most significant canonical pathway was the PID AP1 pathway. The most significant KEGG pathway in hypertrophy was found to be fluid shear stress and atherosclerosis. The terms are ordered on the basis of the log10(q) value (adjusted log10 *p*-value). The list of genes in each cluster is given in Appendix A.

#### 3.5.2. Dilated

Seventy-three DEGs were analysed for functional annotation in the dilatation stage. The most significantly enriched BP was the cellular response to organic cyclic compounds, followed by negative regulation of the protein modification process. Two Reactome gene sets such as scavenging by Class A receptors and formation of tubulin folding intermediates by CCT/TriC were also significantly enriched. The list of genes in each cluster is shown in Appendix A.

#### 3.5.3. LV Failure

A total of 256 DEGs were analysed for functional annotation in the failure stage. The enrichment analysis showed that muscle structure development was the most significantly enriched BP. Extracellular matrix organisation in the Reactome gene set database was significantly enriched. The complete list of genes in each cluster is given in Appendix A.

#### 3.5.4. Recovery

The recovery stage had only five genes found as DEG compared to control, and that did not retrieve any significant enrichment to any pathways.

### 3.6. Association Analysis of DEGs to Diseases

The DEGs in each stage were enriched in the DisGeNET ontology category to find the association of genes with diseases.

#### 3.6.1. Hypertrophy

In the hypertrophy stage, the most significant enrichment was calculated for endotoxemia (log10(q) = −8.7), followed by anoxia. Myocardial ischemia was significantly enriched with 24 genes. The significantly associated diseases with genes in the hypertrophy stage are provided in Appendix A.

#### 3.6.2. Dilatation

In the dilated stage, the most significant enrichment was in myocardial ischemia. There were 14 genes in the dilated group that enriched myocardial ischemia. The other major diseases enriched by dilated DEGs are comedone, stable angina, and hypercholesterolemia. The significantly associated diseases with genes in the dilated stage are provided in Appendix A.

#### 3.6.3. LV Failure

In LV failure, the most significant enriched was familial idiopathic cardiomyopathy (33 genes), followed by vascular diseases (28 genes), and middle cerebral artery occlusion (26 genes). The significantly related diseases with genes in the failure stage are provided in Appendix A.

#### 3.6.4. Novel Potential Candidate Genes in Disease Stages

We performed disease annotation to the DEGs in each stage using DisGeNET. In the hypertrophy stage, the DEGs most significantly enriched juvenile arthritis. However, the highest number of genes in the hypertrophy stage was associated with experimental liver cirrhosis. In the dilated stage, the most significant enrichment was found with myocardial ischemia. The highest number of genes in the dilated stage enriched peptic ulcer. In the failure stage, experimental liver cirrhosis was the most significant disease associated with DEGs (Figure 5A–C). The DEGs in the hypertrophy stage that significantly enriched myocardial ischemia, juvenile arthritis, congenital heart disease, and rheumatoid arthritis are provided in Figure 5A(i–iv). In the dilated stage, the genes that are significantly enriched myocardial ischemia, hypertensive disease, reperfusion injury, and left ventricular hypertrophy are provided in Figure 5B(i–iv). In the failure stage, 21 DEGs were directly related to heart failure (Figure 5C(i)). DEGs expressed in the failure stage related to coronary artery disease, myofibrillar myopathy, and cardiomegaly are given in Figure 5C(ii–iv). Thereafter, the transcripts that were not previously reported to be associated with any cardiovascular diseases were filtered from DEGs of each stage. Figure 5D represents the heatmap of those genes that are potential candidate genes for future cardiovascular research. In the hypertrophy stage, *OTUD1* (OUT deubiquitinase 1), *ARID5A* (AT-rich interaction domain 5A), and *CNTFR* (ciliary neurotrophic factor receptor) are possible targets of further study. In the dilated stage, *TMEM* (transmembrane protein 205), *ADIRF* (adipogenesis regulatory factor), and *NGRN* (neugrin, neurite-outgrowth-associated) can be potential targets. In the failure stage, *ANKRD2* (ankyrin repeat domain 2), *STK38L* (serine/threonine-kinase-38-like), and *ACOT9* (acyl-CoA thioesterase 9) are possible novel targets.

## 4. Discussion

We studied the transcriptome of LV at each stage progressing to HF by RNA-seq. The insertion of an inflatable cuff around the aorta created a pressure overload in the LV. Gradual inflation of the band increased the pressure gradient across the aorta, thus transitioning the LV into distinct stages of hypertrophy, dilatation, and failure. We reached a stage of recovery after the removal of the band. RNA-seq showed that each stage exhibited a distinct set of differentially expressed genes as well as a commonly shared set of genes with other stages. Relieving LV from pressure overload improved LV function into recovery, and this normalised the gene expression. 

Functional annotation by PANTHER database showed similarities in enrichment for BP, MF, and CC in the hypertrophy, dilated, and failure stages. The DEGs in hypertrophy and failure stages have similarly enriched GnRH pathway, which is known for cardiovascular pathogenesis [27]. Likewise, the gene-enriched GnRH pathway may be playing a role in the initiation and progression of cardiac hypertrophy. The other significantly enriched pathways in hypertrophy were PDGF, CCKR, and inflammation by chemokine and cytokine signalling pathways. In contrast to hypertrophy and LV failure stages, the most significant pathways enriched in the dilated stage were inflammation by chemokine and cytokine signalling pathway, followed by integrin, GnRH, and CCKR signalling pathways. The genes that enriched the GnRH pathway in pathological stages of HF were also implicated in cardiomyopathies, and their regulation in the hypertrophic stage of the disease appeared to be a response to alleviate the effect of hypertrophic cues in LV. The hypertrophy-specific genes such as *DUSP1* [28], *NR4A1* [29], and *IRS2* [30] showed a significant increase. The role of *DUSP1* in protecting the heart from cardiomyopathy by inhibiting the MAPK effectors was shown in the Dusp1/Dusp4 double-knockout mouse model. Overexpression of NR4A1 and its nuclear translocation was shown to inhibit isoproterenol-induced cardiomyocyte hypertrophy. Likewise, adenovector-mediated cardiac overexpression of NR4A1 in mice inhibited isoproterenol-induced cardiac hypertrophy. The insulin receptor substrate 2 (IRS2) knockout model is prone to transverse aortic-constriction-induced LV dysfunction. Therefore, it may be reasonable to infer that the increase in these genes may be a response of the myocardium to counteract hypertrophy and fibrosis. The hypertrophy, dilated, and failure stages share the GnRH pathway, CCKR pathway, and inflammation by the chemokine and cytokine signalling pathways in common, although the gene sets enriched these pathways depending on the stage. Further studies are required to elucidate the role of such genes and their related pathways in the development of HF. Cholecystokinin (CCK) in the CCKR pathway is a gut peptide neurotransmitter hormone and is also produced by the heart. The role of CCK has been reported in earlier studies of HF [31]. However, it is not clear as to whether the activation mechanism of CCKR is pathogenic or compensatory, and this should warrant further observations and studies.

Functional annotation using Metascape for GO BP and pathways resulted in a unique enrichment profile for each stage. The hypertrophy stage showed the highest significance for circadian rhythm and angiogenesis. It is well documented that disturbances in circadian rhythm would adversely affect normal cardiovascular physiology [32]. Of the 17 genes that enriched circadian rhythm, most of them are already known to be involved in cardiovascular pathology. However, among those genes that were not reported to be involved in heart pathology is *CIART*, which makes it an ideal candidate for further studies. Angiogenesis was enriched by 27 genes in the hypertrophy stage that are involved in increasing the oxygen supply to the myocardium [33,34]. The significant enrichment of BP and pathways in dilatation and HF stages such as cellular response to organic cyclic compounds, negative regulation of protein modification process, muscle structure development, blood vessel morphogenesis, and apoptotic signalling pathway are broader in classification, and therefore relating these terms as specific to these stages may be improper, even though there may be a contribution from these processes and pathways to HF.

Each disease stage had a unique set of DEGs along with a common set of DEGs presented in other stages. The DisGeNET database provided the list of genes in each stage that are associated with cardiovascular disease. The genes that were not previously reported for cardiovascular disease were selected from the DEG list from each stage and verified by PubMed search for search terms of cardiovascular disease. We avoided genes less than 1.5 log_2_FC. In our model, using this approach, a metastasis suppressor gene, *OTUD1*, was retrieved as a highly expressed gene with 3.4 log_2_FC in the hypertrophy stage. However, it remains to be studied as to whether this is a causative gene in cardiac hypertrophy. Another highly expressed candidate gene, *ARID5A* (2.1 log_2_FC), was reported to stabilise the IL-6 mRNA induced by activation of β-AR in cardiac fibroblast to incite inflammation [35]. Here, we found that the circadian gene *CIART* might be a potential protective gene activated in response to hypertrophic cues, as circadian rhythm controls cardiovascular homeostatic processes [36,37]. Recently, a study using rasagiline mesylate, a monoamine oxidase B inhibitor, prevented post-myocardial infarct LV remodelling by activating circadian gene expression [38]. On the other hand, *NRARP,* a *NOTCH* target gene, was found differentially expressed in hypertrophy samples. The *NOTCH* signalling pathway is important in cardiac development and disease by activating transcription of several genes, such as *EphrinB* and *NRARP* [39,40,41]. *MUSTN1* is mostly found in skeletal muscles and is known to be associated with muscle hypertrophy. Some studies have shown its expression in the heart, but not much is known about whether it contributes to cardiac hypertrophy. *DZANK1* is reported for copy number variant duplication in transposition in great arteries [42]. Single-nucleotide polymorphism in *FBXL17* gene in Lithuanian families has been correlated with coronary heart disease earlier [43]. The role of other genes such as *NARS, PKDCC, ARHGDIG, STARD10,* and *MAPK12* expressed in the hypertrophy stage requires more investigation in cardiovascular disease.

In the dilated stage, *TMEM205, ADIRF, C6H4orf48, NGRN, PROSER1, ERICH2*, and *CINP* were not previously linked to cardiovascular diseases. In contrast, some genes in the possible candidate gene list of the failure stage have been previously implicated in some cardiac pathology. In the failure stage, eight potential candidate genes were identified as upregulated, and nine as downregulated. For example, the *ANKRD2* gene, which is known to play a role in carcinogenesis and myogenesis, exhibited an increased expression in human DCM as a titin-binding protein [44]. Another example is the gene LITAF, which regulates the calcium ion channels in the heart [45]. Overexpression of *LITAF* in rabbit cardiomyocytes decreases the expression of Ca_V_1.2 (subunit of L-type voltage-dependent calcium channel), thereby shortening the duration of the action potential. An intergenic variant of *METTL11B* is found in atrial fibrillation [46]. The other genes in the possible candidate gene list such as *FKBP10*, *OLFML3*, *SKP-1*, *PIGZ*, *PACRG*, *INSYN1*, *DQA*, *CHST2*, *CITED1*, *SBK2*, and *RBFOX3* have been highly significantly regulated in the failure stage. All in all, this study identified potential novel target genes in cardiac hypertrophy, dilatation, and HF, and a systematic approach is needed to ascertain their role in these pathological stages.

## 5. Conclusions

This is a reproducible prospective model of clinical HF that allowed us to sample the myocardium at different pathological stages and study the transcriptome by RNA-seq. Physiological and biochemical recovery of LV was associated with the normalisation of gene expression. Phenotypic changes and functional aberration by pressure overload in LV are associated with pathological expression of genes. The identification of prospective pathological genes improves our understanding of HF pathophysiology and its management.

## Figures and Tables

**Figure 1 biomolecules-12-00731-f001:**
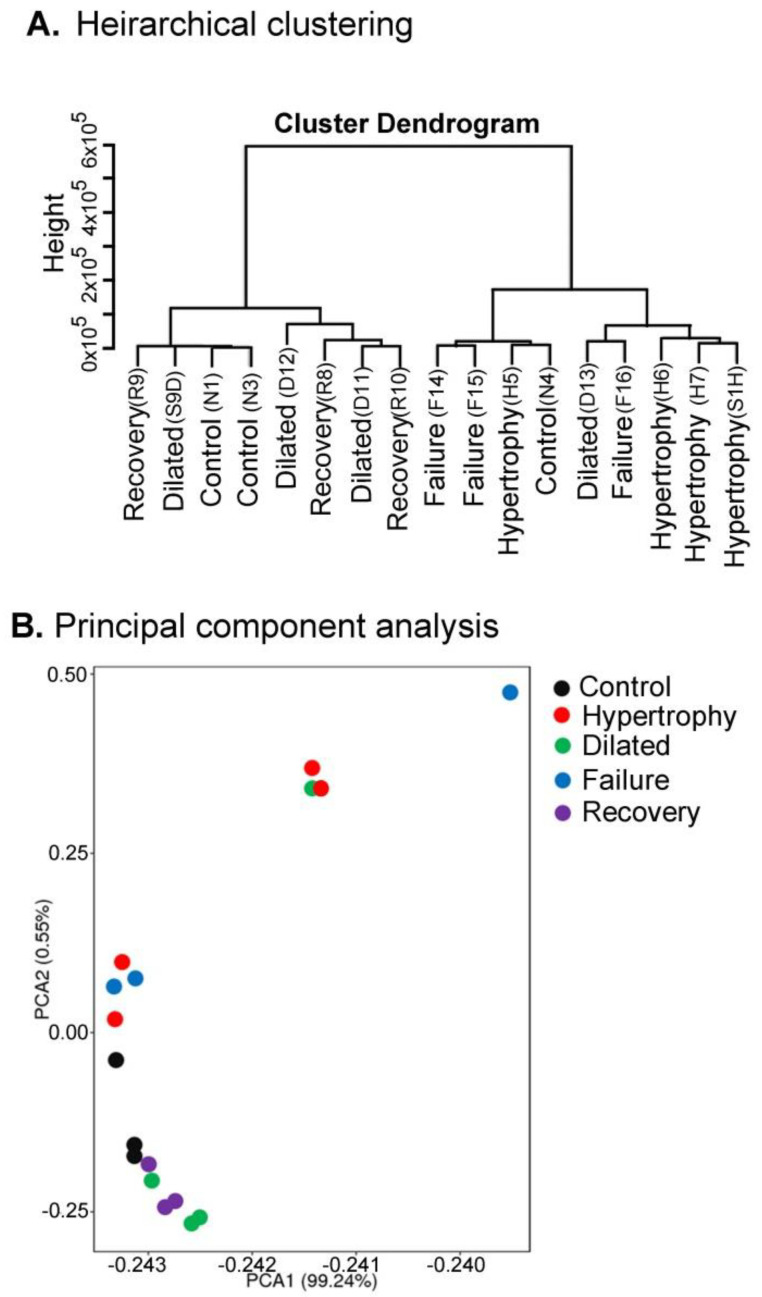
Hierarchical clustering and principal components analysis of gene expression. (**A**) The dendrogram shows hierarchical clustering of all samples according to their gene expression. The closeness of any two samples in the dendrogram shows that samples are similar in their gene expression. (**B**) The principal component analysis shows the samples plotted according to differential gene expression. The distance between points approximates differences in gene expression among samples.

**Figure 2 biomolecules-12-00731-f002:**
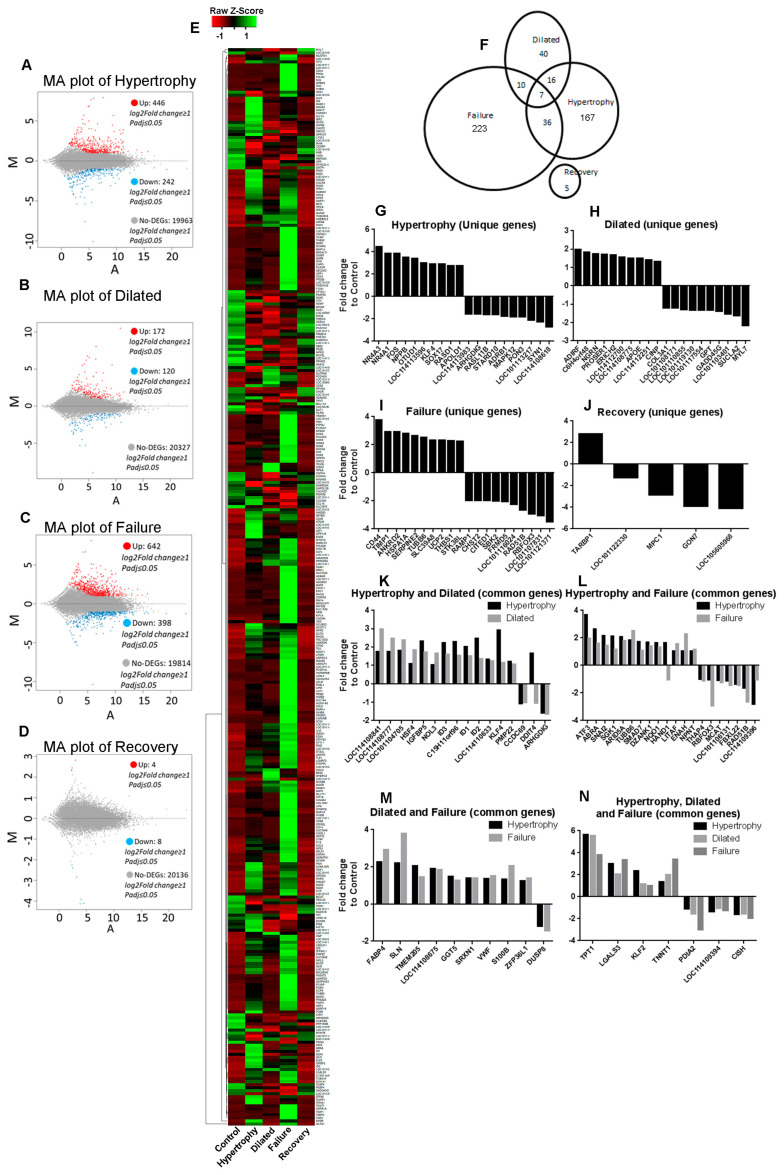
Characteristics of differential gene expression. (**A**–**D**) The MA plots of the hypertrophy through recovery compared to control show the total expression of genes. The X-axis represents value A (log2-transformed mean expression level of each gene). The Y-axis represents value M (log2-transformed fold change of each gene). Red dots represent upregulated DEGs. Blue dots represent downregulated DEGs. Gray points represent genes that are not differentially expressed. (**E**) Heat map showing the differentially expressed genes in each group. (**F**) The Venn diagram shows the distribution of DEGs between the groups. (**G**–**N**) Top up-/downregulated genes that are either unique to a group or common to two or more groups are shown.

**Figure 3 biomolecules-12-00731-f003:**
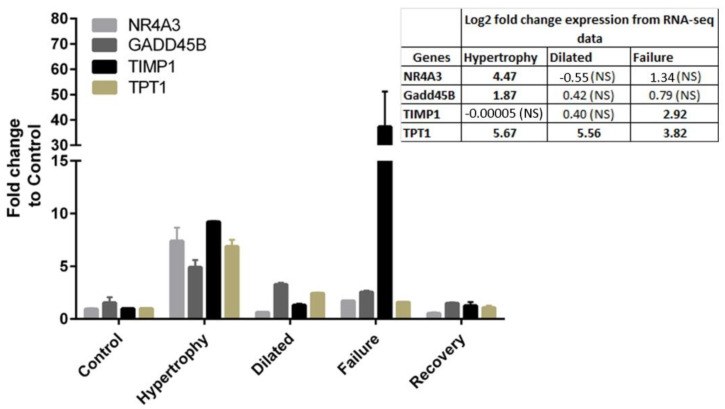
Validation of gene expression by real-time qRT-PCR analysis. The expression of four genes was analysed by qRT-PCR as GAPDH as the housekeeping control gene. The inset table shows the fold change values in the RNA-seq data. NS; non significant.

**Figure 4 biomolecules-12-00731-f004:**
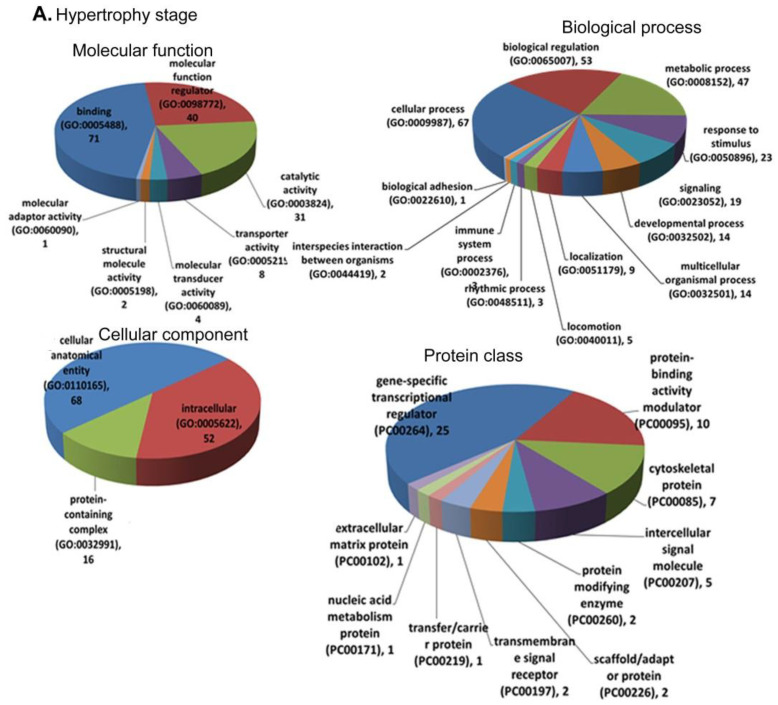
The PANTHER analysis of DEGs for functional annotation. The DEGs of each pathological stage were analysed for molecular function (MF), biological process (BP), cellular component (CC), protein classification (PC), and PANTHER pathways. Representation of the enrichment of each GO term by DEGs of (**A**) hypertrophy stage, (**B**) dilated stage, and (**C**) failure stage. (**D**) The significant PANTHER pathways enriched by DEGs of hypertrophy, dilated, or failure stages.

**Figure 5 biomolecules-12-00731-f005:**
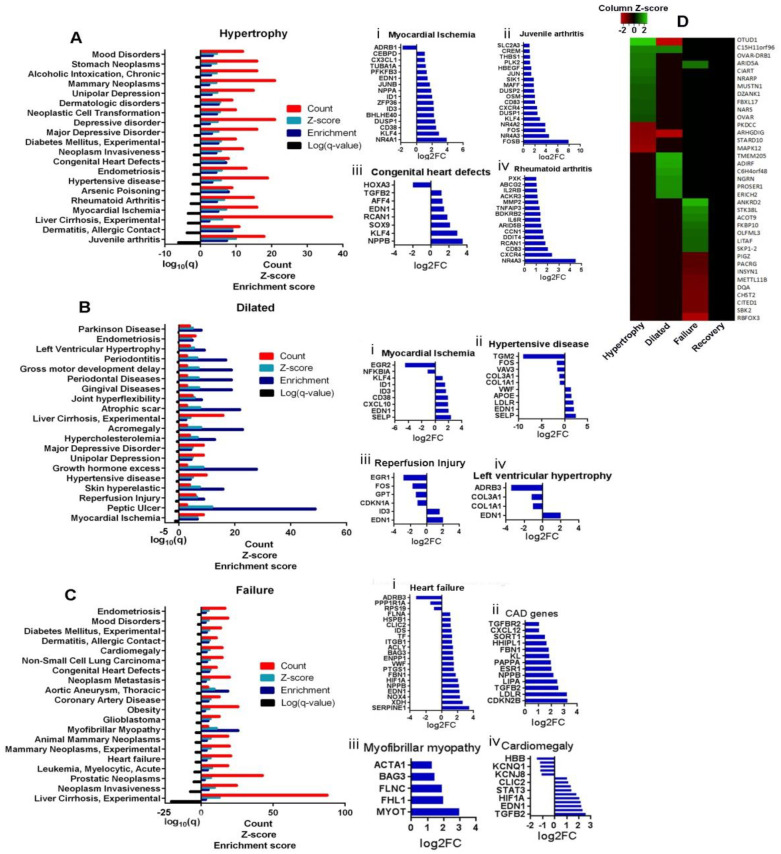
DisGeNET analysis for DEG–disease network identification. The DEGs in each stage were analysed for diseases enriched by them. (**A**–**C**) The graph denotes the significance log10(q) value, enrichment score, z-score, and the count of genes in the stage enriching the disease. (**A**–**C**) (**i**–**iv**)) Subsets of graphs show the DEG-enriched selected diseases. (**D**) The potential candidate genes for cardiovascular diseases were filtered and depicted as a heatmap according to their fold change expression compared to the control.

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
