# Peer review of "Transcriptomal Insights of Heart Failure from Normality to Recovery"

_biomolecules, 2022, doi:10.3390/biom12050731_

Round 1

Reviewer 1 Report

The authors used an NIDCM ovine model to compare left ventricular gene expression changes throughout various stages of heart failure and back to recovery. This is an interesting and important subject and may shed more light onto the molecular events specific for each stage. My major concern with the manuscript is the discordance between the design of the study described in the Methods section and the results shown. I am also missing a figure/graph displaying gene expression levels of selected genes over the 5 stages (in a chronological order). Minor points concern the figures and textual issues.

Major points:

In the Methods section, the authors state that 15 animals were included in the study and 13 animals were analyzed throughout. LV biopsies were collected from all animals at the 5 different stages (Control, Hypertrophy, Dilation, Failure, Recovery). I am therefore expecting to see RNAseq results on 13 x 5 = 65 biopsies. However, results are shown for only 17 biopsies?

I am missing a figure that gives an overview of the gene expression results in a chronological order from control to recovery. Gene-groups could be selected for this based on their differential expression between stages or based on their scientific interest (not yet related to cardiac disease).

Minor points:

1

Although the overall quality of the English language is good, some corrections should be made for better understanding. Examples: Lines 18-20 “Endomyocardial biopsies…” In this sentence “following” should be changed to “followed by”. Otherwise the sentence means that failure follows after recovery. Lines 395-397; “enriched” should be changed to “enriching” and “depending” should be changed to “depended” to better understand the sentence.

2

Lines 144-145: I don’t think the authors mean to state that 22,467 genes were differentially regulated.

3

Section 2.4: I don’t understand why the authors use different FPKM cut-offs for the control and the disease stage groups.

The sentence concerning data not presented in the article should be deleted (lines 175-176)

4

Figure 1:

The results of the hierarchical clustering are not discussed.

It is not clearly written nor discussed which genes were used. I assume that Figure 1A used all genes and Figure 1B only the differential genes? This is not specified in the legend.

In the legend it is not clear to me what the authors mean by “..which can directly reflect the relationship between every two samples.”

5

Figure 2:

Figure 2E should be changed. The color legend is not readable and neither are the gene symbols.

There seems to be an error: Figures 2L and 2M are identical.

6

Lines 262-266. The 7 genes that are common to all three disease stages are subtracted from the common genes between two stages. This is not correct, they should be added (leading to 23, 43 and 17 genes common to two stages). The authors can than state that of these genes, 7 were common to all three disease stages.

7

Figure 3:

The RNAseq results are not shown in the figure, therefore it is not possible to validate the qPCR results from this figure.

8

Line 301: According to Figure 4D, “(14 genes)” should be “(5 genes)”.

9

Lines 325-326. I do not understand the statement “as there are 20 most significant pathway clusters”.

10

Lines 391-395. The authors should discuss why DUSP1, NR4A1 and IRS2 may have a preventive effect on hypertrophy and fibrosis and why the opposite may be true for JUN, ID2, ID3 and ATF3.

Author Response

Reviewer 1 Comments

The authors used an NIDCM ovine model to compare left ventricular gene expression changes throughout various stages of heart failure and back to recovery. This is an interesting and important subject and may shed more light onto the molecular events specific for each stage. My major concern with the manuscript is the discordance between the design of the study described in the Methods section and the results shown. I am also missing a figure/graph displaying gene expression levels of selected genes over the 5 stages (in a chronological order). Minor points concern the figures and textual issues.

Response to Reviewers

Major points:

  1. In the Methods section, the authors state that 15 animals were included in the study and 13 animals were analyzed throughout. LV biopsies were collected from all animals at the 5 different stages (Control, Hypertrophy, Dilation, Failure, Recovery). I am therefore expecting to see RNAseq results on 13 x 5 = 65 biopsies. However, results are shown for only 17 biopsies?

Response: We thank you for your guidance and important comments, and we would like to clarify the sampling from these animals. We used 15 animals in this study and all animals underwent surgical procedures during pathological stages. We collected biopsy several times from all the 13 animals at each stage following confirmation by echocardiography assessment. Taking samples from the endo-myocardial layers of the left ventricle was challenging. The size of each sample was very small. Each sample weighed as little as 0.7 mg and no more than 1.9 mg. We yielded an average between 8-12 samples per procedure per sheep. Having said that, we used quite a lot of biopsy samples for histology, real Time Rt-PCR, immuno Blots and zymography for our previous published work. For this study, the RNA isolation from very small samples was challenging as well. We used the number of samples that provided enough intact RNA yield, free from contaminants, and as highly concentrated as possible. However, we had issues with the RNA isolation from minute quantities of biopsy sample. Therefore, at our best, we selected 3 samples of control, 4 from hypertrophy stage, 4 dilatation stage, 3 heart failure stage and 3 recovery stage from the isolated RNA which were of good quality and quantity.

  1. I am missing a figure that gives an overview of the gene expression results in a chronological order from control to recovery. Gene-groups could be selected for this based on their differential expression between stages or based on their scientific interest (not yet related to cardiac disease).

Response: We agree with the reviewer that a graphical representation of a set of genes based on some criteria, such as genes which are differentially regulated but not previously reported in relation to cardiovascular disease, would be informative. We have given the table of genes as supplementary in this regard. However, accepting the constructive comment of the reviewer, we added a graphical representation of potential candidate genes for cardiovascular disease in each stage as a new figure for this manuscript (figure 5).

 Minor points:

  1. Although the overall quality of the English language is good, some corrections should be made for better understanding. Examples: Lines 18-20 “Endomyocardial biopsies…” In this sentence “following” should be changed to “followed by”. Otherwise the sentence means that failure follows after recovery. Lines 395-397; “enriched” should be changed to “enriching” and “depending” should be changed to “depended” to better understand the sentence.

Response: As per reviewer’s comment the changes were made.

  1. Lines 144-145: I don’t think the authors mean to state that 22,467 genes were differentially regulated.

Response:Thank you bringing this mistake to our attention; it should be a total of 22,467 genes were detected in all samples.

  1. Section 2.4: I don’t understand why the authors use different FPKM cut-offs for the control and the disease stage groups.

Response: We wanted to include as many genes detected in the control group as a baseline to compare with pathological groups.  If we only considered the genes in the control group with a cut-off of FPKM as 5 or more, many of the basal level expressing genes or quiescent genes in the control would have been omitted and that would skew the data. By keeping the cut-off of 5 for FPKM, we will be considering only genes which are significantly expressed in the pathological stages to compare with the control.

  1. The sentence concerning data not presented in the article should be deleted (lines 175-176).

As per reviewer’s comment the sentence is deleted.

  1. Figure 1: The results of the hierarchical clustering are not discussed.

It is not clearly written nor discussed which genes were used. I assume that Figure 1A used all genes and Figure 1B only the differential genes? This is not specified in the legend.

Response: For the dendrogram, the expressions of all the genes were used. And for the PCA analysis, the differential gene expression was used. It is now specified in the legend.

In the legend it is not clear to me what the authors mean by “..which can directly reflect the relationship between every two samples.”

Response: The legend was appropriately changed for more clarification.

  1. Figure 2: Figure 2E should be changed. The color legend is not readable and neither are the gene symbols.

There seems to be an error: Figures 2L and 2M are identical.

Response: A new heatmap was replaced the old one for better reading. 2L and 2M were graphs same and changed accordingly. Thank you noting that.

  1. Lines 262-266. The 7 genes that are common to all three disease stages are subtracted from the common genes between two stages. This is not correct, they should be added (leading to 23, 43 and 17 genes common to two stages). The authors can than state that of these genes, 7 were common to all three disease stages.

Response: I agree with the reviewer that the number of genes common between two stages, say for example between dilated and failure is 17, and of which 7 are common to hypertrophy stage also. The text was changed accordingly.

  1. Figure 3: The RNAseq results are not shown in the figure, therefore it is not possible to validate the qPCR results from this figure.

Response: The fold change of those genes derived from the RNA-seq is provided in the figure as an inset table, so that it can directly be validated.

  1. Line 301: According to Figure 4D, “(14 genes)” should be “(5 genes)”.

Changed accordingly

  1. Lines 325-326. I do not understand the statement “as there are 20 most significant pathway clusters”.

Response: That was syntax error and corrected accordingly.

  1. Lines 391-395. The authors should discuss why DUSP1, NR4A1 and IRS2 may have a preventive effect on hypertrophy and fibrosis and why the opposite may be true for JUN, ID2, ID3 and ATF3.

Response: Based on the appropriate references, the roles of those genes were substantiated. And those genes do not have solid reference to substantiate my statement was omitted. Thank you for correcting that.

Reviewer 2 Report

In this manuscript, Dr. Mohammed Quttainah and colleagues provide insight into the transcriptome in heart failure. Overall, this is a quite nicely written article with a scientific relevance. However, some issues need to be addressed by the authors:

  1. Line 152: The version of "R" should be specified.
  2. Significance should not be reported by just the p value, but also by the CI, since the p value could be ambiguous. An appropriate reading on this can be found here: https://doi.org/10.1038/d41586-019-00857-9
  3. Figure 4: Pie charts could be better arranged in order to enhance and enlarge the writing, as it is is hardly readable

Author Response

Reviewer 2 Comments

In this manuscript, Dr. Mohammed Quttainah and colleagues provide insight into the transcriptome in heart failure. Overall, this is a quite nicely written article with a scientific relevance. However, some issues need to be addressed by the authors:

1.Line 152: The version of "R" should be specified.

Response: The bioinformatics analysis was performed by BGI HongKong, and we filtered the analyzed data for removing the feebly expressing genes and selecting promising candidate genes for further studies. Unfortunately, we are sorry to admit that, their report does not specify which version of R package was used for the analysis.

2.Significance should not be reported by just the p value, but also by the CI, since the p value could be ambiguous. An appropriate reading on this can be found here: https://doi.org/10.1038/d41586-019-00857-9.

Response: As mentioned in the first response, the data was provided for DEG analysis was calculated for p-value adjusted by BGI. They did not provide the CI.

  1. Figure 4: Pie charts could be better arranged in order to enhance and enlarge the writing, as it is is hardly readable

Response: Thank you for your critical comment. Now we have appropriately changed the graph for better readability.

We really are thanking the reviewers for their critical and constructive comments that helped to substantially improve the manuscript. We hope that we addressed the concerns raised by the reviewers as best as we could. We expect that the manuscript is now appropriately revised and suitable to be accepted. Thank you

Round 2

Reviewer 1 Report

My comments on the improved, revised version are the following:

I now understand why only 17 biopsies were used and not 65. This changes the impact of the results somewhat. There seems to be no longitudinal analysis per animal. The authors should indicate whether some samples used for the RNAseq analysis came from the same animal (and if this is the case, they should identify those sample.

Line 255: The deletion of the word “were” makes no sense.

Figure 1A is still not discussed in the text. Therefore, I don’t see a reason for showing it.

Figure 3:

  • According to the y-axis, results shown are Fold Change to control. If this is true, all values for control samples should be equal to 1.
  • The RNAseq results should not be displayed as an inset but should be incorporated in the qPCR results to allow for better comparison.
  • The order of the genes in the color-code legend should be the same as the order of the genes on the X-axis to allow for easier interpretation.
  • Why are there missing values in the RNAseq results?

Line 359: “14” was changed to “4”. This should be “5”.

Paragraph 3.6.4:

There are many spelling errors in this paragraph (experiemntal, enrichemnt, dialted).

Figure 5:

The order of the criteria in the color-code legend should be the same as the order in the figure (y axis).

Author Response

1. In response to the query about sample size and RNA isolation, I have added a new text to the manuscript "lines 142-145" that most probably clarify the message.

2. An extensive grammar check was also conducted, and typos were corrected.

3. Fig 1A was described in detail in the result section.

4. In the qPCR result, the controls are set as one. There was a great variation in one of the control samples (biological replicate) for gene GADD45B. Here there are three biological replicates for each stage. In the control, only one sample of the three biological replicates (which also runs in technical triplicate) is considered as control while calculating the delta delta CT and fold change. Most of the time control biological samples gave comparable CT values and closer to fold change as 1, but rarely in some cases one of the control samples varies a lot for a particular gene of interest, and that skews the value more than or less than one and that is shown by the SD.  

The four genes which are validated by qPCR are not significantly expressed in all stages, they are insignificant in some stages only. That is why those boxes in the table are left blank. Also, the RNA-seq fold change was calculated considering the control as zero, unlike in qPCR the control value is one. So,  making a graph with RNA-seq values will have "no-bars" for some stages. I have changed the table appropriately to reflect that. The color code has been arranged accordingly. 

5. Number "4" changed to "5". Thank you

6. Color code of the graphs in figure 5 has arranged in order.